# Imbalance between Expression of FOXC2 and Its lncRNA in Lymphedema-Distichiasis Caused by Frameshift Mutations

**DOI:** 10.3390/genes12050650

**Published:** 2021-04-27

**Authors:** Sara Missaglia, Daniela Tavian, Sandro Michelini, Paolo Enrico Maltese, Andrea Bonanomi, Matteo Bertelli

**Affiliations:** 1Laboratory of Cellular Biochemistry and Molecular Biology, CRIBENS, Università Cattolica del Sacro Cuore, 20145 Milan, Italy; sara.missaglia@unicatt.it (S.M.); daniela.tavian@unicatt.it (D.T.); 2Psychology Department, Università Cattolica del Sacro Cuore, 20123 Milan, Italy; 3Vascular Diagnostics and Rehabilitation Service, Marino Hospital, ASL Roma 6, 00047 Marino, Italy; sandro.michelini@aslroma6.it; 4MAGI’S Lab, 38068 Rovereto, Italy; matteo.bertelli@assomagi.org; 5Department of Statistical Sciences, Università Cattolica del Sacro Cuore, 20123 Milan, Italy; andrea.bonanomi@unicatt.it; 6MAGI Euregio, 39100 Bolzano, Italy

**Keywords:** Lymphedema-distichiasis syndrome, FOXC2, lncRNA FOXC2-AS1, gene expression, confocal analysis, nuclear aggregates

## Abstract

Forkhead-box C2 (FOXC2) is a transcription factor involved in lymphatic system development. *FOXC2* mutations cause Lymphedema-distichiasis syndrome (LD). Recently, a natural antisense was identified, called lncRNA FOXC2-AS1, which increases *FOXC2* mRNA stability. No studies have evaluated *FOXC2* and FOXC2-AS1 blood expression in LD and healthy subjects. Here, we show that *FOXC2* and FOXC-AS1 expression levels were similar in both controls and patients, and a significantly higher amount of both RNAs was observed in females. A positive correlation between *FOXC2* and FOXC2-AS1 expression was found in both controls and patients, excluding those with frameshift mutations. In these patients, the FOXC2-AS1/*FOXC2* ratio was about 1:1, while it was higher in controls and patients carrying other types of mutations. The overexpression or silencing of FOXC2-AS1 determined a significant increase or reduction in FOXC2 wild-type and frameshift mutant proteins, respectively. Moreover, confocal and bioinformatic analysis revealed that these variations caused the formation of nuclear proteins aggregates also involving DNA. In conclusion, patients with frameshift mutations presented lower values of the FOXC2-AS1/*FOXC2* ratio, due to a decrease in FOXC2-AS1 expression. The imbalance between *FOXC2* mRNA and its lncRNA could represent a molecular mechanism to reduce the amount of FOXC2 misfolded proteins, protecting cells from damage.

## 1. Introduction

Lymphedema-distichiasis syndrome (LD, OMIM 153400) is an autosomal dominant rare disorder in which the lymphatic system, responsible for the production and transport of fluids and immune cells throughout the body, does not develop correctly [1]. LD is characterized by the swelling of the limbs, in particular the legs and feet, and by the presence of extra eyelashes (distichiasis) on the inner lining of both the upper and lower eyelids. Distichiasis can often cause astigmatism or cornea scarring. In addition, swollen and knotted veins, drooping eyelids (ptosis), cardiac anomalies, and cleft palate can be observed [2]. The age of clinical symptoms onset is highly variable in LD patients [3]. 

LD is associated with mutations in the *FOXC2* gene (MIM 602402), a member of the Forkhead-box (FOX) family [4,5]. *FOXC2* is located on the long arm of chromosome 16, positive strand, and consists of a single exon. It encodes a transcription factor (FOXC2), composed of 501 amino acids, regulating the development of some systems during embryogenesis, particularly the lymphatic and blood vascular system. FOXC2 has several functional regions: two transactivation domains, AD-1 and AD-2, located in the N- and C- terminal parts, respectively; a forkhead DNA binding domain (FHD, residues 71–162), which contains the first nuclear localization signal (NLS1, residues 78–93); a second NLS (NLS2, residues 168–176); and an inhibition domain (ID2) located in the C- terminal region [6]. In addition, there are some phosphorylation and SUMOylation sites in the central part of the protein (residues 163–394) [7,8]. 

Almost 110 LD-causing mutations have been identified (www.hgmd.cf.ac.uk accessed on 15 July 2020). Most of them, being complex rearrangements, deletions, insertions, or nonsense mutations, should result in a partial or total loss of FOXC2 function. However, functional studies have shown that some variations, particularly missense mutations, located outside the FHD domain lead to an increase in transcription factor activity [9,10]. These results seem to indicate that LD onset is associated with both loss and gain of FOXC2 function [9,10,11]. To date, the molecular mechanism underlying this pathology is not understood. Moreover, the biological role of some molecules, which could differently regulate FOXC2 expression in LD patients compared to healthy subjects, still needs to be investigated.

Recently, an antisense lncRNA, named lncRNA FOXC2-AS1, was identified [12]. This lncRNA is transcribed by a gene located on the negative strand of chromosome 16. lncRNA FOXC2-AS1 overlaps *FOXC2* mRNA between nucleotides 280–426, establishing a double-stranded RNA structure (Figure 1A). 

This structure allows the stabilization of the *FOXC2* transcript. The direct interaction between *FOXC2* mRNA and FOXC2-AS1 was demonstrated, for the first time, in human doxorubicin-resistant osteosarcoma cell lines [12]. Studies performed in different osteosarcoma cell lines and tissue samples have shown that the expression levels of both *FOXC2* and FOXC2-AS1 were upregulated and that FOXC2-AS1 controlled FOXC2 protein production, stabilizing its transcript. 

In this study, we evaluated the expression levels of *FOXC2* and FOXC2-AS1 in blood samples from some control subjects and LD patients, presenting different *FOXC2* mutations. Moreover, we co-transfected HeLa cells with control and mutant GFP-FOXC2 recombinant plasmids, as well as with FOXC2-AS1 plasmid or FOXC2-AS1 siRNA, to induce significant changes in FOXC2 wild-type and mutant protein synthesis. Finally, we made confocal and bioinformatic analysis to clarify the effect of FOXC2 variations, in particular frameshift mutations, on cellular localization and three-dimensional structure.

## 2. Materials and Methods 

### 2.1. Sample Collection

Peripheral blood specimens were collected from 7 LD patients (3 males and 4 females) and 8 healthy age-matched subjects (4 males and 4 females), using ethylenediaminetetraacetic acid (EDTA) as anticoagulant. The clinical and genetic information of the LD patients were reported in previous studies [13,14]. The LD patients carried different *FOXC2* mutations, reported in Table 1. All subjects involved in the study signed an informed consent agreement. The analyses were authorized by the local Institutional Review Board and performed in agreement with the ethical principles outlined in the Declaration of Helsinki.

### 2.2. Real Time PCR Analysis of FOXC2 and FOXC2-AS1 RNAs Obtained from Blood Cells 

Peripheral blood mononuclear cells (PBMCs) were isolated from the EDTA-treated peripheral blood by density gradient centrifugation using Histopaque, following the manufacturer’s instructions. RNA was extracted with Trizol solution and reverse transcription was performed after treating 2 μg of RNA with DNAse I [15]. Quantitative real time PCR (qRT-PCR) was made with 50 ng of cDNA in a 20 μL-reaction volume using SsoAdvanced Universal SYBR Green Supermix (Biorad) and the primers FOXC2-F: AGTGCAGCATGCGAGCGATG; FOXC2-R: CGAGAGGGCCTCGTCCAGG; FOXC2-AS1-F: CGAGAGGGCCTCGTCCAGG; FOXC2-AS1-R: TTGCCTTCTAGTCGCCTCC. qRT-PCR protocol was as follow: 35 cycles at 95 °C for 15 s, 54 °C for 2 min [16]. Analyses were realized in triplicate and the relative quantification was estimated using the 2^−ΔΔCt^ method, after normalization with Glyceraldehyde-3-Phosphate Dehydrogenase (GAPDH) expression. RNA obtained from HeLa cells was used as positive control.

In addition, semi-quantitative RT-PCR was performed as previously described [17], using the following primers: FOXC2-2F CGCCCGAGAAGAAGATCAC; FOXC2-2R CGCTCTTGATCACCACCTTC; FOXC2-AS1-2F CTTGCCGGGCTTCTTGTCGT; FOXC2-AS1-2R TACATTTTCGTCTTCTGTTCTTTTATTGG. To exclude DNA contamination, FOXC2 expression analysis was carried out on samples with and without reverse transcriptase (+RT and −RT). GAPDH level was used to normalize FOXC2 and FOXC-AS1 concentration in each sample. Finally, amplification products were sequenced (Eurofins Genomics, DNA sequencing service).

### 2.3. Localization of Mutant FOXC2 and Evaluation of Protein Expression in HeLa Transfected Cells

GFP-FOXC2 recombinant plasmids were previously obtained [10,18]. FOXC2-AS1 cDNA was amplified from HeLa cells, then subcloned into pcDNA3.3-TOPO (Life Technologies) and finally sequenced (Eurofins Genomics, DNA sequencing service). siRNA FOXC2-AS1 and scramble oligonucleotide (negative control) were purchased from Eurofins Genomics. FOXC2 and FOXC2-AS1 recombinant plasmids and siRNAs were transfected into HeLa cells using Lipofectamine 3000 (Thermo Fisher Scientific, Waltham, MA, USA), following the manufacturer’s instructions. 

After 48 h, the cells were lysed using NP40 buffer containing protease inhibitors. Total proteins were quantified using a Pierce^TM^ Coomassie (Bradford) Protein Assay Kit (Thermo Fisher Scientific, Waltham, MA, USA) and separated by electrophoresis on 10% SDS-polyacrylamide gel (Biorad, Hercules, CA, USA). Then, they were transferred to a polyvinylidene difluoride (PVDF) membrane (Biorad) and, after blocking with 5% nonfat dry milk, the membrane was incubated with primary antibodies against GFP (1:5000, ABNOVA, Taipei, Taiwan) and GAPDH (1:5000, ABNOVA). The immune signals were examined using the SuperSignal West Pico Complete Detection Kit (Pierce) containing ImmunoPure Peroxidase Conjugated Goat anti-Mouse (dil 1:20,000). 

In addition, HeLa cells were simultaneously transfected as described above, then fixed and analyzed with a LeicaMB5000B microscope and a confocal microscope Leica TCS SPE equipped with 40× and 63× oil immersion objectives. 

### 2.4. Statistical and Bioinformatic Analysis

SPSS v.19 package (SPSS, Chicago, IL, USA) was utilized to carry out all statistical analysis. qRT-PCR results were tested comparing the values with Student’s *t*-test (*p*-value ≤ 0.05 was statistically significant). Linear correlations were performed using the Pearson product moment method (significant set at *p* < 0.05). The ratio of FOXC2-AS1/*FOXC2* expression was compared using the Kruskal–Wallis test (*p*-value < 0.05 was considered to indicate significance). 

The investigation of the secondary and three-dimensional structure of FOXC2 mutant proteins was performed using Robetta software (https://robetta.bakerlab.org accessed on 15 July 2020). 

## 3. Results

### 3.1. FOXC2 and lncRNA FOXC2-AS1 Expression Levels in Peripheral Blood

The patients enrolled in this study presented different *FOXC2* variants: three missense mutations (p.I213V, p.V228M and p.A492V), a nonsense mutation (p.Y109*), and two frameshift mutations (p.H199Pfs264* and p.I213Tfs18*). These were localized in various protein domains (Figure 1B), differently effecting FOXC2 function. Missense mutations caused either a partial loss or a significant gain of FOXC2 activity, while nonsense and frameshift mutations determined a complete or severe loss of transcriptional function, respectively (Table 1) [10,18].

To evaluate the expression levels of FOXC2 and FOXC2-AS1, total RNA obtained from peripheral blood of healthy subjects, LD patients and HeLa cells was analyzed using qRT-PCR assay (Figure 2A). The results showed no significant difference of *FOXC2* and FOXC2-AS1 expression between these two groups of subjects. Pearson correlation analysis showed that *FOXC2* and FOXC2-AS1 expression was strongly correlated in healthy subjects (R = 0.878, *p* < 0.05), while a moderate correlation was identified in the LD patients (R = 0.577, *p* < 0.05) (Figure 2B,C). Nevertheless, dividing the LD patients in two groups, those with frameshift and those with other *FOXC2* mutations, a strong positive correlation between *FOXC2* and FOXC2-AS1 RNA amount could also be found in the second group of subjects (R = 0.926, *p* ≤ 0.001) (Figure 2D). Finally, in patients carrying frameshift mutations, it was observed that the ratio of FOXC2-AS1/*FOXC2* expression tended to 1:1 and it was significantly lower (*p* < 0.05) than that detected in control subjects and all the other patients (Figure 2E). 

Further evaluations revealed that *FOXC2* and FOXC2-AS1 expression was significantly higher in the female population compared to male population (Figure 3A). Moreover, *FOXC2* and FOXC2-AS1 RNA amount positively correlated in males (R = 0.767, *p* < 0.05), but not in females (R = 0.263, *p* > 0.05) (Figure 3B), consistently with the fact that all the LD patients with frameshift mutations were females. 

To definitively ascertain if *FOXC2* and FOXC2-AS1 were expressed in blood cells, a semi-quantitative RT-PCR analysis was carried out in the control and LD patients. *FOXC2* and FOXC2-AS1 RT-PCR fragments of 384 and 319 nt, respectively, were obtained. No *FOXC2* amplification products were observed in -RT samples. The sequencing of *FOXC2* and FOXC2-AS1 RT-PCR products confirmed the expression of both genes in peripheral blood (Appendix A).

### 3.2. Modulation of FOXC2 Protein Synthesis through Up- or Down-Regulation of FOXC2-AS1.

To verify whether the modulation of FOXC2-AS1 expression could influence both FOXC2 wild-type and mutant protein production, we co-transfected HeLa cells with wild-type or mutant GFP-FOXC2 plasmids and with FOXC2-AS1 or FOXC2-AS1 siRNA. After co-transfection, Western blot and immunofluorescence analyses were performed. 

Western blot analysis displayed that FOXC2-AS1 overexpression increased FOXC2 wild-type and mutant protein levels (Figure 4A, lanes 4 and 8). In contrast, endogenous FOXC2-AS1 knockdown dramatically reduced the amount of FOXC2 proteins (Figure 4A, lanes 5 and 9).

Immunofluorescence investigation supported immunoblot results. Indeed, the co-transfection of GFP-FOXC2 wild-type or mutant plasmids with FOXC2-AS1 enhanced the production of recombinant proteins. On the contrary, co-transfection with FOXC2-AS1 siRNA decreased FOXC2 proteins’ expression (Figure 4B).

### 3.3. Immunofluorescence and in Silico Evaluation of FOXC2 Frameshift Mutations

We previously showed that all FOXC2 mutations but one (p.Y109*) did not affect protein nuclear localization [10,18].

Immunofluorescence analysis evidenced that p.Y109* showed a cytoplasmic localization, missense mutations (p.I213V, p.V228M and p.A492V) maintained a homogeneous nuclear distribution, similar to native FOXC2, and frameshift mutations H199Pfs264* and I213Tfs18* were able to enter the nucleus and induced intranuclear protein aggregates (Appendix A). 

To better clarify if these nuclear aggregates could interact with DNA, confocal microscope analysis was performed. The nuclear images of cells transfected with GFP-FOXC2(H199Pfs264*) or GFP-FOXC2(I213Tfs18*) clearly showed an uneven DNA staining which colocalized with mutant FOXC2 protein aggregates (Figure 5).

*FOXC2* frameshift mutations cause the production of mutant proteins (p.H199Pfs264* and p.I213Tfs18*) in which a stretch of amino acids can be found that is different from native FOXC2 sequence. Bioinformatic investigation revealed secondary structure modifications not only of amino acids localized in the central region but also in some residues of the FHD domain (Appendix A). In particular, both predicted structures revealed alterations of N86 (from coiled coil to α-helix), L125-C128 (from coiled coil to α-helix), P150-D151 (from α-helix to coiled coil), F161 (from coiled coil to α-helix), and L162 (from coiled coil to β-strand in p.H199Pfs264*and from coiled coil to α-helix in p.I213Tfs18*). Three-dimensional models displayed a dramatic modification of both tertiary structures caused by conformational changes of FHD domain and the loss of more than 50% of the wild-type sequences (Appendix A).

## 4. Discussion

In this explorative study, the RNA expression level of *FOXC2* and FOXC2-AS1 was evaluated in healthy subjects and LD patients. First, we demonstrated that both genes were expressed in blood cells, although at very low levels. We failed to find a significant difference between the healthy subjects and LD patients, while the analysis of the two transcripts showed differences between males and females. In particular, higher levels of both RNAs were detected in females. These data are in agreement with genetic and epigenetic studies that demonstrated sex-specific regulation of some genes [18,19,20]. Recently, it was also shown a diverse expression of some lncRNAs between males and females [21,22]. The different regulation of genes and lncRNAs expression can protect or predispose males and females against disease onset in different manners. LD involves equally males and females but arises in males at an earlier age than females [4,23]. The sex difference of *FOXC2* and FOXC2-AS1 expression levels might explain later lymphedema onset in females. Higher levels of *FOXC2* and FOXC2-AS1 RNA should lead to an increase in the number of FOXC2 proteins in females. Therefore, the presence of a higher amount of FOXC2 molecules might temporarily compensate for transcriptional activity impairment in females carrying *FOXC2* mutations, delaying lymphedema onset.

Pearson’s analysis highlighted that *FOXC2* and FOXC2-AS1 RNA levels were strongly correlated in healthy subjects and in patients, excluding those with frameshift mutations (Appendix A). Indeed, the ratio of FOXC2-AS1/*FOXC2* RNA indicated that there was an imbalance in the expression of the transcription factor and its lncRNA in patients carrying frameshift mutations. Many studies reported that alteration of lncRNA expression can lead to a wide variety of disorders. Both upregulation and downregulation of lncRNAs involved in many physiological processes, such as MALAT1, PLUT, and CARMN, are associated with the onset of various cancers, as well as metabolic, cardiovascular, and neurological disorders [24,25,26]. In LD, the dysregulation of lncRNA expression might have a different biological meaning. In the presence of specific *FOXC2* variations, such as frameshift mutations, the decrease in FOXC2-AS1 expression might contribute to the reduction in FOXC2 mutant proteins production. This mechanism might represent a way by which the cell protects itself.

In our study, we have demonstrated that both overexpression and silencing of FOXC2-AS1 can modulate the production of wild-type and mutant FOXC2 proteins. 

As it was previously reported, p.I213V, p.V228M, and p.A492V preserved the correct nuclear localization, while could cause both a loss (p.V228M) and gain (p.I213V and p.A492V) of FOXC2 activity [10,18]. p.Y109* lost the ability to localize into the nucleus and, consequently, did not retain transcriptional function [18]. Finally, the frameshift mutations, such as H199Pfs264* and I213Tfs18*, maintained nuclear localization but generated abnormal protein aggregates [18]. Moreover, they severely impaired FOXC2 activity and also prevented cellular proliferation. We provided evidence that cells transfected with GFP-*FOXC2*(H199Pfs264*) and GFP-*FOXC2*(I213Tfs18*) lost their ability to divide, increasing cell death [18]. It is unknown how *FOXC2* frameshift mutations might block mitosis. However, it might be hypothesized that nuclear protein aggregations play an important role in the inhibition of cell cycle progression. Here, we demonstrated by confocal analysis that these aggregates colocalized with DNA, showing a non-uniform staining. As chromatin assembly is essential to regulate transcription factor-DNA interaction and gene expression [27], the variation of DNA distribution might have a deleterious impact on cell viability and proliferation. 

In silico investigation showed that nuclear aggregates formation could be associated with dramatic structural alterations of FOXC2 mutant proteins. In the previous study, we investigated secondary and tertiary structures of both mutant proteins using i-Tasser tool [18]. i-Tasser is based on identification of PDB templates through Local Meta-Threading Server. Data obtained showed a higher number of amino acids involved in an α-helix structure, particularly in FHD domain, and dramatic alterations of tridimensional conformation. To verify the accuracy of i-Tasser prediction, in this study, we used another bioinformatic tool based on Rosetta macromolecular modelling suite, which provides both ab initio and comparative models of protein domains. Structural alterations reported by this software are similar to those highlighted by i-Tasser. Indeed, the analysis predicted misfolded protein generation for both p.H199Pfs264* and p.I213Tfs18*. These data are in agreement with the observations of Zhang and colleagues who hypothesized that the modification of FOXC2 C-terminal part might determine a decrease in protein stabilization causing an abnormal three-dimensional structure [23]. Interestingly, our analysis showed conformational alterations also of some FHD amino acids, with an increase in residues presenting an α-helix structure. FHD is the region which regulates DNA-FOXC2 interaction and the amino acids, which directly participate in DNA binding, are involved in an α-helix secondary structure [28,29]. Our data indicate that the mutant proteins might be able to strongly bind DNA, carrying modifications of chromatin conformation. 

## 5. Conclusions

In conclusion, we showed that there were similar expression levels of *FOXC2* and FOXC2-AS1 in blood samples from control and LD subjects. Conversely, a sex-based regulation of both genes was identified. Moreover, a strong positive correlation between both RNAs amount was identified in all participants, except in females carrying *FOXC2* frameshift mutations. In these LD patients, FOXC2-AS1 amount was lower than in other control and LD subjects, determining an imbalance between *FOXC2* and FOXC2-AS1 expression. Confocal and bioinformatic investigations seemed to indicate that frameshift mutations dramatically modify FOXC2 tertiary structure and DNA binding. Our in vitro studies showed that ectopic FOXC2-AS1 over or under expression directly regulates FOXC2 protein production. Likewise, endogenous FOXC2-AS1 down-regulation might be a molecular mechanism useful to protect the cell, decreasing misfolded proteins production. Since a small number of blood samples were analyzed, additional studies should be performed on larger number of healthy and LD subjects to better clarify the relationship between *FOXC2* and FOXC2-AS1 expression.

## Figures and Tables

**Figure 1 genes-12-00650-f001:**
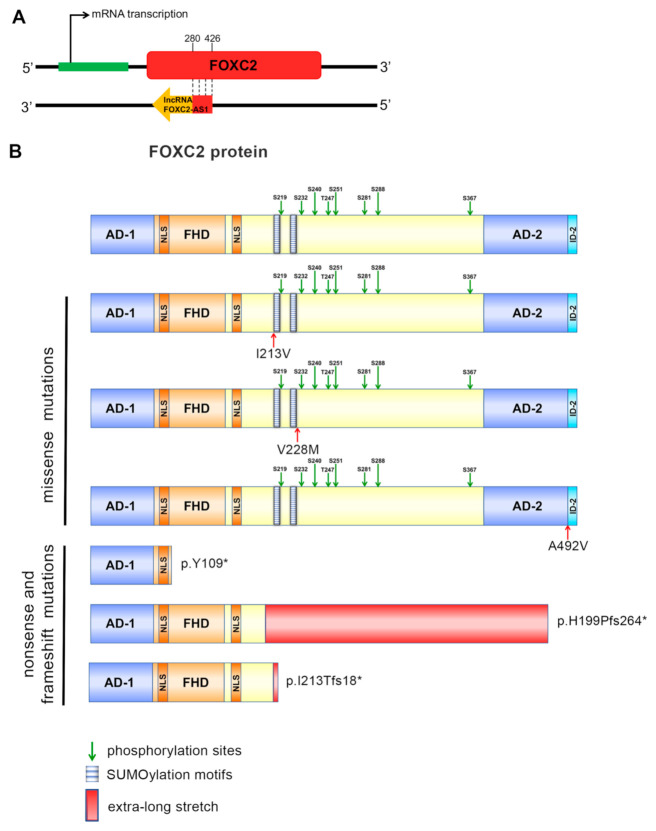
Representation of *FOXC2* mRNA and lncRNA FOXC2-AS1 interaction, and of FOXC2 protein structure. (**A**) Antisense lncRNA is transcribed from the complementary strand of *FOXC2* gene and it forms a double-stranded RNA through binding with *FOXC2* mRNA between nucleotides 280-426. (**B**) Schematic representation of structural domains of FOXC2 protein: Activation Domains (AD-1, amino acids 1–70, and AD-2, amino acids 395–494); Forkhead Domain, the DNA-binding region (FHD, amino acids 71–162); Nuclear Signals (NLS1, amino acids 78–93, and NLS2, amino acids 168–176); the central region (amino acids 163–394) with two SUMOylation motifs and eight phosphorylation sites; Inhibitory Domain 2 (ID-2, amino acids 494–501). Moreover, a schematic representation of FOXC2 mutant proteins is also reported.

**Figure 2 genes-12-00650-f002:**
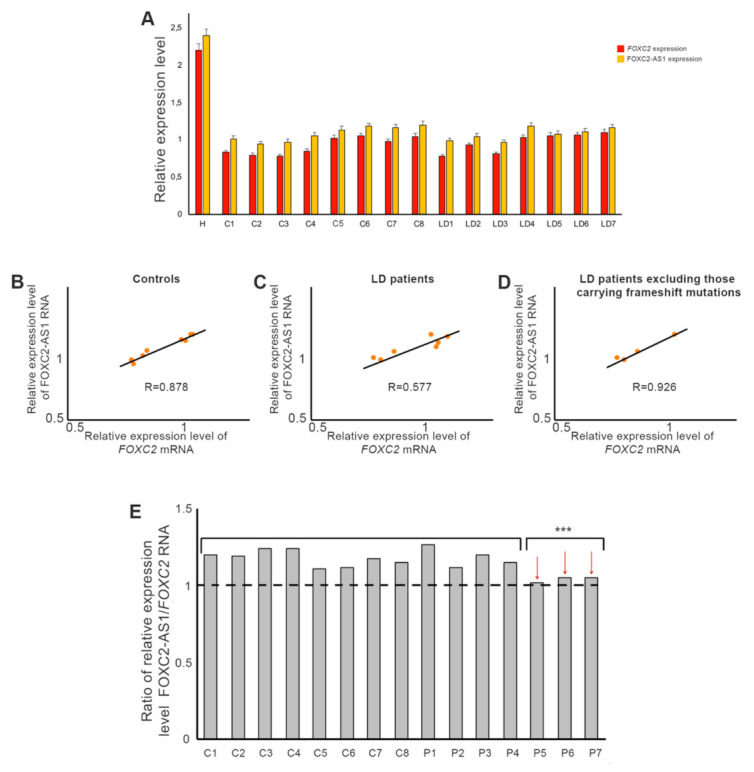
*FOXC2* and lncRNA FOXC2-AS1 expression in control and LD blood samples. (**A**) Real time PCR analysis of *FOXC2* and lncRNA FOXC2-AS1 RNA in healthy (C1–C8) and LD (P1–P7) subjects. HeLa cells (H) were used as positive controls. *GAPDH* was used as a housekeeping gene to normalize the expression levels. Each bar represents mean of three independent experiments. (**B**) Pearson correlation analysis between *FOXC2* and FOXC2-AS1 expression levels observed in control subjects (correlation coefficient R = 0.878, *p* < 0.05), in all LD patients (**C**) (correlation coefficient R = 0.577, *p* < 0.05) and in LD patients carrying *FOXC2* missense and nonsense mutations (**D**) (correlation coefficient R = 0.926, *p* ≤ 0.001). (**E**) Ratio of lncRNA FOXC2-AS1 and *FOXC2* expression levels in controls and LD subjects. Patients carrying frameshift mutations were indicated by arrows. Statistical analysis was performed using Kruskal–Wallis test. *** *p* < 0.05.

**Figure 3 genes-12-00650-f003:**
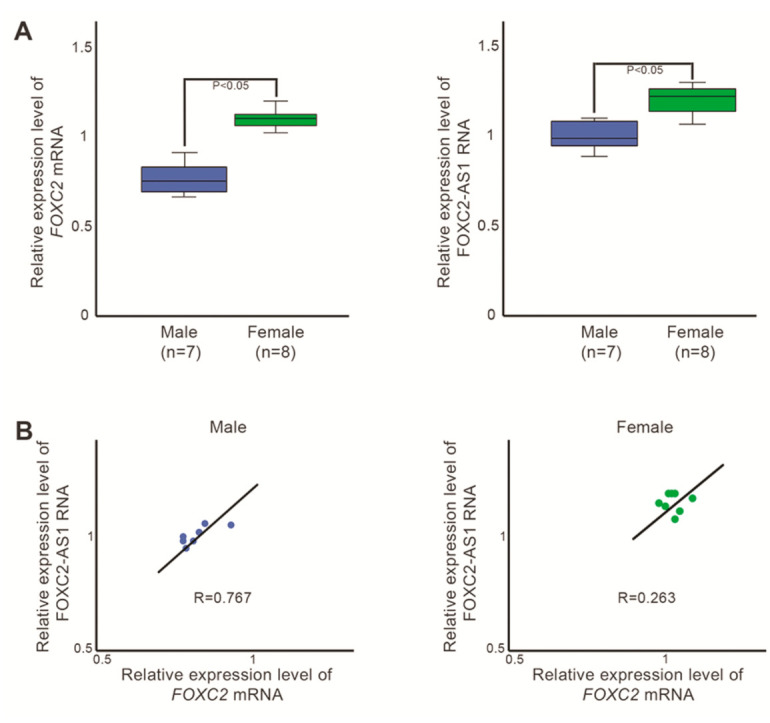
Evaluation of *FOXC2* and lncRNA FOXC2-AS1 expression profile in male and female blood samples. (**A**) Box plots of *FOXC2* (left) and FOXC2-AS1 (right) expression values in males (control and LD) and females (control and LD), detected by RT-PCR. Statistical analysis was assessed using Student’s *t*-test. *p*-values of < 0.05 was considered significant. (**B**) Pearson correlation analysis between *FOXC2* and FOXC2-AS1 expression levels observed in males (control and LD) and females (control and LD). Male correlation coefficient: R = 0.767 (*p* < 0.05). Female correlation coefficient: R = 0.263 (*p* > 0.05).

**Figure 4 genes-12-00650-f004:**
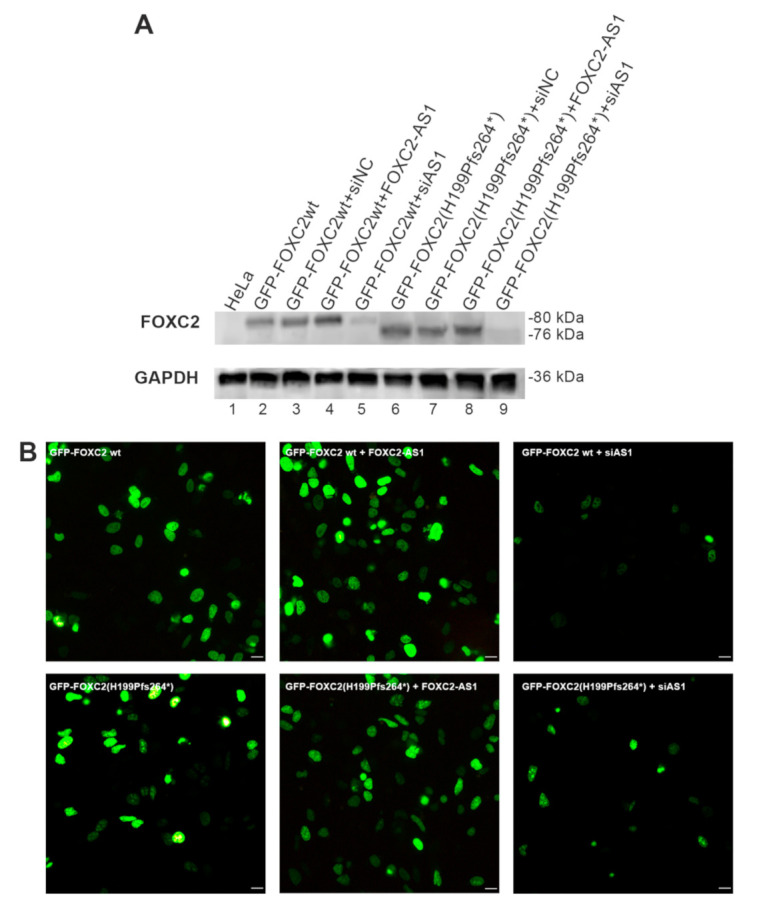
Analysis of FOXC2 wild-type and mutant proteins after FOXC2-AS1 overexpression or silencing. (**A**) Western blot analysis of FOXC2 protein expression in HeLa cells co-transfected with different expression vectors and siRNAs (GFP-FOXC2wt; GFP-FOXC2wt + siNC; GFP-FOXC2wt + FOXC2-AS1; GFP-FOXC2wt + siAS1; GFP-FOXC2(H199Pfs264*); GFP-FOXC2(H199Pfs264*) + siNC; GFP-FOXC2(H199Pfs264*) + FOXC2-AS1; GFP-FOXC2(H199Pfs264*) + siAS1). Lane 1: non-transfected HeLa cells. (**B**) Immunofluorescence analysis of HeLa cells after transfection with GFP-FOXC2wt; GFP-FOXC2wt + FOXC2-AS1; GFP-FOXC2wt + siAS1; GFP-FOXC2(H199Pfs264*); GFP-FOXC2(H199Pfs264*) + FOXC2-AS1; GFP-FOXC2(H199Pfs264*) + siAS1. After 48 h, the cells were fixed with 3% paraformaldehyde and FOXC2 recombinant proteins were observed by direct immunofluorescence analysis of GFP tag (in green); 20× magnification. Scale bar: 10 μm.

**Figure 5 genes-12-00650-f005:**
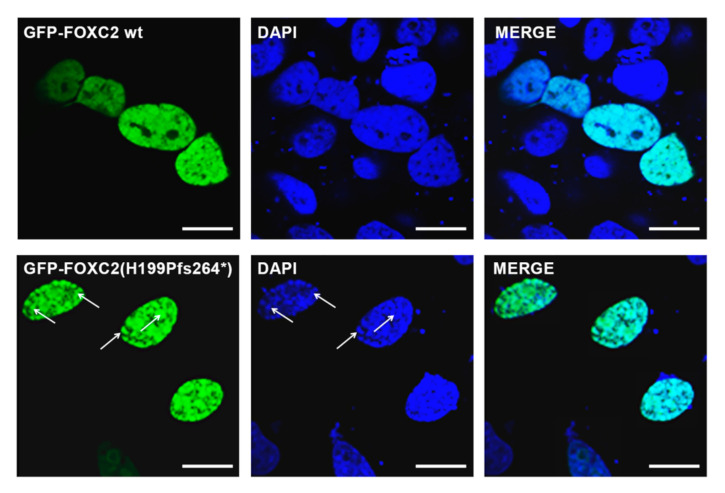
Confocal analysis of HeLa cells transiently transfected with GFP-*FOXC2* and GFP-*FOXC2*(H199Pfs264*) constructs. The evaluation of protein nuclear localization reveals that p.H199Pfs264* generates protein aggregates which colocalize with an irregular DNA staining. In green, FOXC2 constructs tagged with GFP. In blue, nuclei stained with DAPI. Magnification: 40×. Scale bar: 10 μm. Putative regions of colocalization between FOXC2 mutant protein and DNA are indicated by arrows.

**Table 1 genes-12-00650-t001:** *FOXC2* mutations identified in LD patients enrolled for the study.

Patient	Mutation	Protein	Transcriptional Activity of Mutant Protein *
P1 (M)	c.1475C>T	p.A492V	182%
P2 (M)	c.682G>A	p.V228M	70%
P3 (M)	c.327C>A	p.Y109*	0%
P4 (F)	c.637A>G	p.I213V	103%
P5 (F)	c.595dupC	p.H199Pfs264*	28%
P6 (F)	c.638delT	p.I213Tfs18*	30%
P7 (F)	c.638delT	p.I213Tfs18*	30%

* Transcriptional activity percentage of FOXC2 mutant proteins was determined considering FOXC2 wild-type signal as 100% of activity (Luciferase reporter assay) [10,18].

## Data Availability

Results of patients’ genetic analysis were reported in Michelini, S. et al, Lymphology, 2012, *45*, 3-12; Michelini, S. et al, Lymphology, 2016, *49*, 57–72. Data of functional studies were reported in Tavian, D et al, 2016, doi:10.18632/oncotarget.9797; Tavian, D.; et al, 2020, doi:10.3390/ijms21145112.

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
