# Peer review of "Imbalance between Expression of FOXC2 and Its lncRNA in Lymphedema-Distichiasis Caused by Frameshift Mutations"

_genes, 2021, doi:10.3390/genes12050650_

Round 1

Reviewer 1 Report

Because human FOXC2 is not expressed in blood cells in databases of Human protein atlas (https://www.proteinatlas.org/ENSG00000176692-FOXC2/blood) and GTEx Portal (https://gtexportal.org/home/gene/FOXC2), I requested FOXC2 expression in blood cells by western blot. But authors did not show the FOXC2 protein expression. I cannot believe in Figure2-3 of qRT-PCR data in blood cells.

Reviewer 2 Report

Thank you very much for all correction. I am recommending paper for publication.

Author Response

We thank reviewer 2 for recommending paper for  publication.

Reviewer 3 Report

The present study aims to enrich our understanding of a rare genetic disease, Lymphedema Distichiasis (LD). LD has been associated with mutations occurring in FOXC2 and is inherited in an autosomal dominant pattern. Several mutations (nonsence, missence and frameshift mutations) have been identified in FOXC2 and associated with LD.

FOXC2 expression has been detected in several tissues. A previous study of the same researchers reports the expression of FOXC2 also in peripheral blood mononuclear cells (PBMCs) of LD patients. The molecular mechanism of how FOXC2 is involved in LD is largely unknown but in 2017 it was reported that in osteosarcoma, a long non-coding RNA (lncRNA - FOXC2.AS1) binds to and stabilizes FOXC2. Several aspects of the disease remain to be addressed, for example, how FOXC2 expression is regulated and why men manifest the disease at an earlier age than women.

In the present study, the authors were interested in investigating whether FOXC2 and its lncRNA FOXC2.AS1 are expressed in the PBMCs of healthy controls (n=8) and LD patients (n=7; 3 men and 4 women). They tested this by Real Time PCR and report that both genes are expressed in PBMCs of healthy controls and patients at a balanced pattern (FOXC2AS.1/FOXC2 >1) except in the presence of frameshift mutations. In the presence of frameshift mutations, FOXC2AS.1/FOXC2 =1. Moreover, they show that FOXC2.AS1 expression is linearly correlated with FOXC2 expression except in the presence of frameshift mutations. These findings strongly suggest that the expression of both genes is tightly regulated to maintain a strict balance. This balance is disrupted by the presence of frameshift mutations in FOXC2.

In order to verify that FOXC2 and FOXC2.AS1 are truly expressed in PBMCs, they sequenced fragments of these genes amplified by semi-quantitative PCR. Furthermore, they report that the expression of both genes is higher in women compared to men. 

Because other studies have shown that FOXC2.AS1 binds to and stabilizes FOXC2 mRNA, the authors investigated if this occurs also in the case of FOXC2 that carries frameshift mutations. They tested this by transfection of HeLa cells with plasmids expressing a GFP-fused wt or mutated (frameshift mutations) FOXC2 along with a plasmid expressing FOXC2.AS1 and siRNAs against FOXC2.AS1. By western blot and confocal microscopy, they show that FOXC2.AS1 is capable of stabilizing FOXC2 even in the presence of frameshift mutations. Moreover, FOXC2 carrying frameshift mutations retains its nuclear localization ability but forms protein aggregates colocalizing with DNA.

Finally, by using a bio-informatic approach, the authors provide prediction of the structural alterations of FOXC2 in the presence of two frameshift mutations.

Suggested novel findings:

  • FOXC2 and FOXC2.AS1 are expressed in PBMCs in a balanced and positively correlated pattern (please see “Major comments: required experiments - graphs, #1”)
  • Frameshift mutations in FOXC2 cause an alteration in the ratio of expression level of FOXC2.AS1/ FOXC2 and disrupt the correlation of expression of FOXC2.AS1 and FOXC2.
  • Higher expression of FOXC2 and of FOXC2.AS1 in the PBMCs of women compared to men
  • Regulation of the expression level of wt FOXC2 as well as mutated (frameshift mutations) form of FOXC2 by FOXC2.AS1, in HeLa cells.
  • Frameshift mutated form of FOXC2 is capable of entering the nucleus, colocalize with DNA and form protein aggregates (please see “Important issue to consider, #1”)
  • Frameshift mutations are predicted to cause major structural alterations in FOXC2.

Major comments: required experiments - graphs:

  1. Since a previous study, using a single cell RNA sequencing approach, has failed to detect FOXC2 expression in PBMCs (Chen et al, 2018), and because the authors believe that FOXC2 is expressed at very low levels in these cells, it is necessary to perform a negative control experiment to verify that the experimental conditions do not result in the amplification of FOXC2 from cells having no endogenous FOXC2 expression. This should be done by following the same experimental steps as those followed in the study (24 cycles and 50ng of input cDNA). A candidate cell type can be the HEK293T cell line (Lidell et al, 2011).

Important issue to consider

  1. Figure 5 and Supplementary Figure S1 showing that a frameshift mutant form of FOXC2 is capable of entering the nucleus, colocalize with DNA and form protein aggregates - similar data have been previously published by the same group (Tavian et al, 2020 – Figure 2C). Since in both studies, the authors have used the same cell line, constructs, examined the cells at the same time after transfection (48hrs) and clearly show that the nuclear aggregates colocalize with DNA (although they don’t say that in the text, they show this with DAPI), what is the difference in the current study? Maybe a higher magnification photo with arrows showing putative regions of colocalization and non-colocalization be more informative. This should be accompanied with appropriate change in the text.

 Minor comments:

  1. Add paragraphs and titles in Materials and Methods section
  2. Correct numbering of Controls in Figure 1A: Control 5 (C5) is missing, and one control has been falsely named as Control 10 (C10).
  3. In line 175, after “HeLa cells” they should add “(H)” so that the sentence becomes “HeLa cells (H) were used as positive controls” in order to show that HeLa cells correspond to the bar named as “H”.
  4. Although the authors have done a similar bioinformatic analysis in their previous study (Tavian et al, 2020 - pp 9-10; Fig 7), they don't mention this in the Discussion section to compare their findings. 
  5. Same axes in Figure 1
  6. List of abbreviations

References

Chen, J., Cheung, F., Shi, R., Zhou, H. and Lu, W., 2018. PBMC fixation and processing for Chromium single-cell RNA sequencing. Journal of translational medicine16(1), pp.1-11.

Lidell, M.E., Seifert, E.L., Westergren, R., Heglind, M., Gowing, A., Sukonina, V., Arani, Z., Itkonen, P., Wallin, S., Westberg, F. and Fernandez-Rodriguez, J., 2011. The adipocyte-expressed forkhead transcription factor Foxc2 regulates metabolism through altered mitochondrial function. Diabetes60(2), pp.427-435.

Tavian, D., Missaglia, S., Maltese, P.E., Michelini, S., Fiorentino, A., Ricci, M., Serrani, R., Walter, M.A. and Bertelli, M., 2016. FOXC2 disease-mutations identified in lymphedema-distichiasis patients cause both loss and gain of protein function. Oncotarget7(34), p.54228.

Round 2

Reviewer 3 Report

The authors have answered all comments and therefore I consider their study adequate to be published in Genes